# Advanced Exergy Analysis in the Dynamic Framework for Assessing Building Thermal Systems

**DOI:** 10.3390/e22010032

**Published:** 2019-12-25

**Authors:** Ana Picallo-Perez, José M Sala, George Tsatsaronis, Saeed Sayadi

**Affiliations:** 1Research Group Energy in Buildings (ENEDI), Department of Thermal Engineering, University of the Basque Country (UPV/EHU), 48013 Bilbao, Spain; josemariapedro.sala@ehu.eus; 2Institute for Energy Engineering, Technische Universität Berlin, 10623 Berlin, Germany; georgios.tsatsaronis@tu-berlin.de (G.T.); s.sayadi@tu-berlin.de (S.S.)

**Keywords:** dynamic advanced exergy analysis, grey box modeling, heating and DHW systems, avoidable and unavoidable exergy destruction, endogenous and exogenous exergy destruction

## Abstract

This work applies the Dynamic Advanced Exergy Analysis (DAEA) to a heating and domestic hot water (DHW) facility supplied by a Stirling engine and a condensing boiler. For the first time, an advanced exergy analysis using dynamic conditions is applied to a building energy system. DAEA provides insights on the components’ exergy destruction (ED) by distinguishing the inefficiencies that can be prevented by improving the quality (avoidable ED) and the ones constrained because of technical limitations (unavoidable ED). ED is related to the inherent inefficiencies of the considered element (endogenous ED) and those coming from the interconnections (exogenous ED). That information cannot be obtained by any other approach. A dynamic calculation within the experimental facility has been performed after a component characterization driven by a new grey-box modelling technique, through TRNSYS and MATLAB. Novel solutions and terms of ED are assessed for the rational implementation of the DAEA in building energy installations. The influence of each component and their interconnections are valuated in terms of exergy destruction for further diagnosis and optimization purposes.

## 1. Introduction

The worldwide awareness for reducing the per capita energy demand is well known since the requirements of the world’s increasing population can lead to unsustainable situations. As reported in Reference [1], if the current trend in energy use remains the same, the demand for oil from 2007 to 2035 is expected to grow by 30% and the demand for coal and natural gas is expected to rise by 50%. Therefore, there is some urgency for lowering the energy consumption, particularly in building energy systems, since nearly one-third of the total primary energy supply in the world is used in the building sector [2], mainly in Heating, Ventilation, and Air Conditioning (HVAC) systems [3]. Because of that, significant energy savings can be achieved through proper operation of those systems [4]. Thus, the development of energy efficient facilities represents a great concern and has become the focus of many research activities [5]. Notwithstanding the potential of these facilities, improvement of their efficiencies is a complex and dynamic issue that needs special treatment.

In order to obtain more useful information from an energy system, an exergy analysis should be conducted in addition to a conventional energy analysis, since the former measures the maximum theoretical useful work that can be obtained [6] by accounting not only for the quantity of energy but also for its quality [7]. Exergy, unlike energy, does not satisfy a conservation law, but it is destroyed when the quality of energy is degraded because of the irreversibilities within different processes [8]. In other words, the idea that something can be destroyed through a process is very useful in the design, analysis, and optimization of energy systems [7]. An energy balance fails to identify the true thermodynamic inefficiencies, and, thereby, an evaluation based exclusively on the energy concept might be misleading.

In a conventional exergy analysis, the exergetic efficiency (ε) is used as an indicator to characterize a component in terms of performance and compare it to similar components in other systems [9]. It is defined as the ratio between the productive purpose (the goal) of the component, known as *Product* ($E_{P}$), and the resources required to achieve that objective, labelled as *Fuel* ($E_{F}$) [10] (both of them expressed in exergy terms). In addition, if no exergy losses are present, the difference between $E_{F}$ and $E_{P}$ is equal to the exergy destruction ($E_{D}$) of the component [11]. However, this conventional exergetic analysis cannot satisfactorily evaluate the mutual interdependencies among the system components, and does not consider the real potential for improving every component [12].

Under these circumstances, to overcome those barriers, the Advanced Exergetic Analysis (AEA) was developed. In this methodology, part of the inefficiencies caused by the component itself and caused by the interactions with other components can be evaluated separately, as well as the fraction, which can be avoided through technological improvements [12,13,14]. Consequently, the main purpose of a plant analysis in the AEA is the identification of the causes of irreversibilities and their effects in terms of $E_{D}$ and in terms of performance [15]. If the reader wants to explore other theories related to the same objective, Reference [16] can be consulted.

As stated above, there are some irreversibilities that every component may have, which cannot be reduced even using the best and current technical alternatives, due to physical, technological, and economic constraints or limitations. They are known as unavoidable exergy destruction ($E_{D}^{UN}$) [17,18], whereas the remaining part is named as avoidable exergy destruction ($E_{D}^{AV}$). Hence, the splitting up of these two inefficiencies gives a realistic picture of the potential to improve the thermodynamic efficiency of each component.

In addition to this, there is another way to distinguish the irreversibilities. Some exergy destructions encountered in a component are due to the inefficiencies within other components. This means that $E_{D}$ occurring in one component of a system does not only depend on its performance, but it is also influenced by the inefficiencies of the remaining components [11]. That part of the $E_{D}$ associated with the exergy destruction in other components is called exogenous exergy destruction ($E_{D}^{EX}$), while the part caused only by internal irreversibities within the component itself is named as endogenous exergy destruction ($E_{D}^{EN}$). Thus, the separation of those two exergy destructions enables us to better understand the interactions of the system components and to consider those interactions in optimization procedures. Nonetheless, the calculation of $E_{D}^{EN}$ and $E_{D}^{EX}$ for a component is more difficult than that of $E_{D}^{UN}$ and $E_{D}^{AV}$. This is a major challenge of AEA.

Subsequently, the avoidable and unavoidable $E_{D}$ can be related to the endogenous and exogenous ones. That information can be used to investigate: (1) which part of the exergy destruction within a component can be decreased by improving the component itself ($E_{D}^{EN.AV}$), (2) which part can be reduced by a structural enhancement of the overall system or by improvements in other components ($E_{D}^{EX.AV}$), and (3) and (4) the exergy destructions, which cannot be prevented (reflected by $E_{D}^{EX.UN}$ and $E_{D}^{EN.UN}$). In consequence, the information acquired by the AEA can be applied for system design, control improvement, and maintenance purposes.

The novel aspects and originality of the present study are associated with the AEA application to a building thermal facility and with proceeding from a steady-state analysis to a dynamic one. Although the subtask A and C of the IEA ECBCS Annex 49 project was to focus on dynamic exergy analysis in buildings [19], the AEA methodology was not expressly used. In this case, we implement it under a dynamic point of view. For this reason, we call the proposed methodology Dynamic Advanced Exergy Analysis (DAEA), which adds some new aspects to the steady-state AEA. Thus, one of the aims of the paper is to present a guideline on how to apply a detailed DAEA.

For instance, Reference [20] applies an AEA to an air conditioning system using average values to evaluate the cooling process during the day and the accumulation process at night. Reference [21] applies AEA to low-exergy analysis for existing building heating systems. In a similar way, Reference [22] makes a comparison between two geothermal district heating systems based on an AEA. Although these papers are related to a buildings framework, none of them (or any other similar paper) considers the dynamic state of the system in the AEA. This makes the present paper, under the authors’ knowledge, the first study of its kind in the application of DAEA.

On the other hand, an AEA has been applied in several research studies to different industrial fields. References [14] and [23] apply this method to combined-cycle power plants. Reference [11] performs a detailed AEA to an absorption refrigeration machine, and Reference [24] examines a gas engine heat pump drying system. An AEA application to cogeneration systems is presented in Reference [25]. Despite the abundance of studies about AEA, all of them have been carried out in a steady-state context.

The work in this paper is presented as follows: initially, the heating and DHW experimental testing facility is presented. Then, the procedure to obtain a grey-box model for the principal components of the facility is explained. Afterward, a dynamic advanced exergetic analysis is conducted, and the calculated avoidable and unavoidable exergy destructions for each component are discussed. The following part focuses on the methodology for calculating the endogenous and exogenous $E_{D}$ in a dynamic context. Moreover, some novel terms such as the endogenous exergy destruction based on the real characteristic curves ($E_{D}^{EN^{*}}$) are introduced.

The numerical results are presented and discussed. Lastly, some comments on the merits and drawbacks of this new DAEA are made.

## 2. Dynamic Representation of the Building

Dynamic characterization of the studied building facility is performed as the first step since the leap from the experimental data to the mathematical model is a key factor for the proper application of DAEA. Every component taking part in the facility must be accurately and dynamically represented in order to represent the reality as accurately as possible.

One of the problems with the detailed energy simulation of a building facility is the fact that many inputs (some may not be practicably measurable) are required for model definition [26]. Therefore, the characterization strongly depends on the availability and quality of the experimental data.

According to Reference [27], two types of models can be defined depending on the gathered data: (1) The law-driven models and (2) the data-driven models. The last require prior data knowledge and use system behavior to predict system properties.

The concerning case follows the second model characteristics since the data are taken from a limited number of sensors and monitoring devices in the system. The model is built based on the grey-box modeling approach, which is well-proven as a comprehensive and accurate method for modeling dynamic systems [28] by identifying certain key parameters using a physical system model. There are some building transient simulation tools based on these premises such as TRNSYS [29], which implements a component-based simulation approach through a modular structure.

Likewise, system identification can be used for parameter estimation in a dynamic model representation. It is used for building models based on the observed behavior of the system so that variables like thermal inertia can be reflected. Consequently, the characterization of the facility under study and the parameter adjustments were achieved by combining the TRNSYS v17 software-specific components with the MATLAB System Identification Toolbox (R2014a) [30].

The representation of complex systems through necessary simplifications and simulation constraints can be questioned since some information could be lost within that transition. As listed in Reference [26], various sources of uncertainty can be found in the system identification, the modeling, the numerical processing, and the scenario itself. In any modeling environment, the quality of the outputs can only be as good as the quality of the available inputs.

## 3. Case Study

This section deals with the detailed description of the methodology, according to the selected case study.

### 3.1. Description of the Facility

The experimental facility of the Laboratory for the Quality Control in Buildings (LQCB) has been used to obtain the needed data. This facility aims to test new energy technologies (that could be incorporated in buildings) in order to obtain useful information that benefits different stakeholders of the building sector. Through this plant, the operation of various types of equipment and pioneering technologies can be analyzed, and different control strategies can be tested in order to reduce fuel consumption and ${\mathrm{CO}}_{2}$ emissions. This facility is based on a flexible concept, so that diverse types of energy conversion equipment can be tested, with some based on fossil fuels and others based on renewable energies [31]. Figure 1 shows the general layout of the building’s thermal facility under study including some of the energy conversion units.

The facility customized for the DAEA application is based on a micro-cogeneration Stirling engine (S) and a condensing boiler (CB) (a Stirling was chosen in order to introduce a ground-breaking component into a building system). That engine supplies 1 kW electricity, and 3.7–5 kW thermal energy from the combustion gases, depending on the operating temperatures and modulation. Additional 20 kW thermal energy can be produced in an auxiliary boiler inside this unit. Another equipment used in the energy conversion layer of the facility is a natural gas condensing boiler which provides 28 kW thermal energy with a manufacturer’s energetic efficiency (based on the lower heating value) of 97%.

These components provide the required thermal energy and DHW equivalent to the demand of three single-family dwellings located in Vitoria (Northern Spain). Additionally, there is a 351 hydraulic compensator (HC), a plate heat exchanger (HX), a 1000 L storage tank (T), distribution pipes, hydraulic pumps, three-way valves (V1, V2, V3), and a fan coil (FC), as shown in Figure 1.

On the one hand, the DHW demand profile was calculated using the Generation of Domestic Hot Water Tool in Reference [32], which creates profiles based on a statistical analysis. Afterward, it was discretized, programmed, and controlled by a high-accuracy flow meter. This profile uses 5-min discrete values, which are continuously compared with the energy data obtained from the temperature and flow meters associated with the facility.

On the other hand, the heating profile was obtained from a TRNSYS simulation. The building envelope characteristics were defined as well as the usage and the climate data, through the TMY2 type. The emulation of the heating demand is done through a fan coil battery together with a three-way valve so that the heating demand profile defined in advance is matched by its modulation and operation of that component.

The control strategy is such that the Stirling engine has priority over the condensing boiler so that it is switched when there is demand for thermal energy or when the average temperature of the DHW tank falls below 60 °C.

The control is managed together with an expansion module and is connected via Ethernet to a PC. The sensors distributed in the facility provide more than 120 signals and, thereby, enable engineers to control the desired variables and to ensure proper operation of the plant. In this regard, 46 temperature sensors Pt 100 are installed with 40 of them on the energy distribution system and the rest on the deposits. Moreover, 11 electromagnetic flow meters with an uncertainty less than 0.1% are installed. There are two pressure switches in the system as well, which function as sensors to measure the ambient pressure and humidity, and some other measuring devices function to record the indoor temperature and humidity inside the building containing the facility. Lastly, two gas meters of Class 1.5 are installed to measure fuel consumption in the condensing boiler and the micro-cogeneration unit, and 1 electric meter of Class 1 is used to record the amount of generated electricity by the Stirling unit.

### 3.2. Selection of the Components

In order to properly conduct a DAEA, the next step requires the proper selection of every subsystem, with respect to their productive final purpose. For instance, if the supply and return collectors located above the hydraulic compensator are considered individually (driving mixer and returning diverter separately), no productive final purpose can be defined for them. Conversely, if they are jointly handled instead (C), the productive purpose of this component would be the coverage of both the DHW and the heating demand. Similarly, the splitter and the mixer of the three-way valve right before the fan coil should be considered inside the same component (V3). Then the goal would be to supply the heating demand. Following those considerations, the components used for the analysis are listed in Table 1, where their names and abbreviations are shown.

### 3.3. Characterization of the Components

Once the components are selected, after collecting the experimental data from the corresponding four-day test, the system was dynamically characterized (with a 5-min time step) by the grey-box modeling technique and via the implementation of TRNSYS and MATLAB. The components were first modeled individually and, then, the entire facility was simulated.

As stated above, the modeling of every component relied on TRNSYS. So, first of all, the appropriate type from its libraries that best fits every component has to be chosen. In such a way, the characterization is performed based on the mathematical reference of the TRNSYS components. Therefore, the required inputs to the software (which matched the actual measured values obtained from the facility) are considered to be independent variables, and, accordingly, the acquired outputs from the TRNSYS simulation are considered dependent ones. For instance, the variable $\left(T_{i_{Dep}}^{Tr}\right)$ obtained from the TRNSYS simulation and the real values measured in the LQCB experimental facility $\left(T_{i_{Dep}}^{Re}\right)$ might have a deviation because TRNSYS does not consider the additional inertia encountered in the components of the experimental facility. That refinement is performed using the MATLAB System Identification Toolbox, by incorporating inertia and the consequent real behavior to adapt the outputs of TRNSYS to reality $\left(T_{i_{Dep}}^{Calc}\right)$. This is explained later in detail.

Figure 2 serves as an example of how the individual characterization of the supply and return collectors (C) and heat exchanger (HX), which have been programmed in the interface Simulation Studio of TRNSYS.

As can be seen, different modules take part in the adaptation and improvement of the models, as explained below.
The component itself, chosen from the TRNSYS library, is located in the middle with its abbreviated name, according to Table 2 (C or HX). First, the type needs to be chosen and then adapted to the characteristics of the LQCB experimental facility. That is done by changing the parameters of the selected TRNSYS Type.The user types, which appear in the component’s surrounding (labeled as $T_{i}^{Re}$), correspond to the external data readers containing the experimental values. In this case, the monitored data from the LQCB experimental facility needs to be inserted, according to the chosen time step. The number of the user types is equal to the number of recorded variables from the sensors.There are two types of user variables, which are the independent $\left(T_{i_{Ind}}^{Re}\right)$ and dependent ones $\left(T_{i_{Dep}}^{Re/Tr/Calc}\right)$, denoted by the subscripts “Ind” and “Dep,” respectively. The independent variables correspond to the inputs of the component being analyzed and are located in the left side of the component in Figure 2, while the dependent user variables that are the actual measured values taken from the test facility are placed on the right side. These dependent variables should be compared to other dependent variables, which resulted from the TRNSYS simulation to obtain a model that is able to describe the real behavior of the component. To sum up, there are three different dependent variables: those acquired from the TRNSYS simulation $\left(T_{i_{Dep}}^{Tr}\right)$, the real ones provided by the sensors in the experimental facility $\left(T_{i_{Dep}}^{Re}\right)$, and the ones calculated in MATLAB $\left(T_{i_{Dep}}^{Calc}\right)$. Since the first two values are not the same $\left(T_{i_{Dep}}^{Tr}\neq T_{i_{Dep}}^{Re}\right)$, a mathematical relationship between the independent values $\left(\sum_{j}T_{j_{Ind}}^{Re}\right)$ and the dependent ones must be found in order to represent reality (see Equation (1)). This step is conducted in the MATLAB System Identification Toolbox.
(1)$$T_{i_{Dep}}^{Calc}=f\left(T_{i_{Dep}}^{Tr},\;\underset{j}{\sum }T_{j_{Ind}}^{Re}\right)=T_{i_{Dep}}^{Re}+error$$For the reason mentioned above, $T_{i_{Dep}}^{Re}$ and $T_{i_{Dep}}^{Tr}$ are connected to the MATLAB Type in Figure 2 (with the parameter identification tag). This type refers to the interconnection between both software where the adjustment of dependent variables is carried out. To achieve that, the experimental data have been compared to the values obtained from TRNSYS, so that the new output from MATLAB is adjusted to the reality.

In this way, a mathematical model of every component of the system can be developed and implemented in the simulation of the entire facility.

### 3.4. Conventional Exergy Analysis

Once the models of the individual components and the overall system are completed (and mass and energy balances are fulfilled), the DAEA can be initiated. The first step is to conduct a conventional exergy analysis in order to calculate the exergetic efficiency and exergy destruction for each component. The definition of the dynamic exergetic efficiency is a delicate step and likely one of the most important ones in completing this study correctly. As cited before, the variable that unambiguously characterizes the performance of a component from the thermodynamic viewpoint is an appropriately defined exergetic efficiency, i.e., the ratio between the product $E_{P}$ and the fuel $E_{F}$ in the studied component [13]. In this context, the Specific Exergy Costing (SPECO) approach [33] is used as a generic methodology to, among other things, identify the exergies of fuel and product ($E_{F}$ and $E_{P}$) for each subsystem. Even in this systematization, the definitions of $E_{F}$ and $E_{P}$ must be carefully considered since special exceptions can occur in dynamic situations.

#### Application to the Experimental Building Thermal Facility

In addition to the previously stated concerns, in the present case, a correct dynamic representation is a key point since the definition of $E_{F}$ and $E_{P}$ in a component can vary over time, depending on the actual productive structure of the component. As an example, different definitions of $E_{F}$ and $E_{P}$ exist for the storage tank T, and, consequently, for its exergetic efficiency $\varepsilon_{T}$, since this depends on which flows are considered to be product and fuel. At any given time, the tank could be just charging and, then, $E_{F}$ would be the incoming exergy supplied to the tank and $E_{P}$ would be the increase of the accumulated exergy within the tank. Otherwise, it might be the case that only DHW demand is activated. Therefore, $E_{F}$ would then be the decrease of the stored exergy within the tank and $E_{P}$ would be the exergy of the DHW leaving the tank, or we could have the situation that charging and discharging go on concurrently and, then, $E_{F}$ is the exergy supplied to the tank plus the decrease of the stored exergy, whereas $E_{P}$ would be the exergy of the DHW leaving the tank. Accordingly, in this phase of the exergetic efficiency analysis, the inertia that arises in the system plays a significant role, since it can alter the $E_{F}$ and $E_{P}$ assessment.

Taking into account the above considerations, a conventional dynamic exergetic analysis can be performed. Accordingly, for all exergy analyses, the choice of the reference temperature is a crucial and important part. Therefore, it is assumed that, for all the components, the system boundaries are at the reference environment dynamic temperature. Thus, there are no exergy losses associated with heat losses from the components [34]. Hence, exergy losses would only appear at the level of the overall facility, whereas, for every component, the difference between $E_{F}$ and $E_{P}$ corresponds to the exergy destruction $E_{D}$ within this component.

### 3.5. Unavoidable and Avoidable Exergy Destructions

The unavoidable exergy destruction is constrained by technological limitations and is calculated considering each component in isolation (i.e., separated from the system) assuming the most favorable operating conditions. These conditions refer to the absolute minimum of exergy destruction when the product remains unchanged. The conditions are associated with very low temperature differences and very small losses of thermal and mechanical exergy within the component being analyzed [14,17]. Nevertheless, the assumptions for simulating unavoidable conditions depend on the engineers’ personal expertise and scope. Therefore, they are arbitrary to some extent.

The avoidable exergy destruction in the *k*th component of the system ($E_{D,k}^{AV}$) is the difference between the total and the unavoidable exergy destructions within the same component (Equation (2)).
(2)$$E_{D,k}^{AV}=E_{D,k}-E_{D,k}^{UN}$$

Having reached this point, it is important to note that, even if the unavoidable and avoidable exergy destructions are individually acquired, the behavior of the remaining components affects the calculation of both exergy destructions. This happens because the independent variable values remain the same as in reality or, in other words, when calculating avoidable exergy destruction. The real input data are enforced to the component with the best attainable technology, and the effect of the remaining components is included in those imposed input data. Consequently, $E_{D,k}^{AV}$ encompasses both the inefficiencies that could be avoided if better quality equipment would be developed and the inefficiencies that can be prevented by avoiding interactions among the components. For this reason, the combinations of $E_{D,k}^{UN\cdot EN}/E_{D,k}^{UN\cdot EX}$ and $E_{D,k}^{AV\cdot EN}/E_{D,k}^{AV\cdot EX}$ must be considered to separate these effects.

#### Application to the Experimental Building Thermal Facility

For calculating $E_{D,k}^{UN}$, the major sources of inefficiency in each component need to be first identified and, afterward, the minimum exergy destruction criterion should be applied. Different groups of components are, hereafter, exposed and listed, according to their level of difficulty.
(C/V1/V2/V3) The exergy destructions in the mixers are mainly caused by mixing streams at different temperatures and pressures. Thus, if the control system acts in a way that equal temperatures and pressures are assumed, the unavoidable exergy destruction becomes zero.(ITF) The aim of this component is to guarantee the best thermal conditions of the inlet flow to the condensing boiler. Hence, all irreversibilities should be avoidable if that flow is already at its nominal state.(FC) The unavoidable exergy destruction within the fan coil is slightly lower than the exergy destruction within the fan coil with the highest energetic efficiency in the market.(HX) The exergy destruction within a heat exchanger can be reduced by: (1) matching streams of similar heat capacity rates, which achieves parallel temperature profiles [35], (2) selecting small temperature differences between the average temperatures of the hot and cold streams, (3) considering an adiabatic heat exchanger, (4) neglecting pressure losses, and (5) choosing the maximum available heat transfer coefficient in each zone of the heat exchanger.(HC/T) The main cause of exergy destruction in those components is the mixing of the high and low-temperature portions of the storage medium. To enhance the storage performance, it is, therefore, necessary to decrease the mixing losses by inserting the heat transfer fluid in the temperature layer, which is closer to the stream temperature. This process entails managing the injection of heat into the corresponding temperature level by a correct stratification [36]. Moreover, it is shown that the exergy storage capacity increases as the degree of stratification arises. The exergy destruction is decreased by augmenting the thermal conductivity of the heat exchanger inside the tank [37]. Additionally, the heat losses are reduced by improving the insulation.

Bearing all that in mind, the conditions assumed for calculating the $E_{D}^{UN}$ in HC and T are set out below: (1) as the inlet and outlet positions of the heat transfer fluids are fixed, the incoming flows cannot be inserted in different specific layers. Nevertheless, the maximum stratification profile should be considered. (2) Due to legal regulations [38], the storage temperature should be greater than 60 °C in order to avoid legionella formation and the same rule also influences the inlet fluid temperature, (3) the DHW charging and discharging depend on the user demand profile, so no optimal heat process periods can be enforced, (4) there is no heat exchanger inside the tank of the LQCB experimental facility, and (5) lastly, thermal insulation should be added to the tank surface, so that it becomes adiabatic.

• (S/CB) Systems including a combustion process are usually, by far, the components with the highest exergy destruction. Hence, a detailed separated exergy analysis should be carried out. The main causes of inefficiencies in a combustion process are friction, mixing, a chemical reaction, and heat transfer. The $E_{D}$ caused by friction is significantly lower than the ones caused by a chemical reaction and heat transfer. The $E_{D}$ due to isobaric mixing depends, once again, on the differences in temperature and chemical composition of the streams that are mixed. That exergy destruction can be sometimes reduced, but it is often entirely unavoidable. The $E_{D}$ resulting from chemical reactions can be reduced by letting the reactions take place closer to their thermodynamic equilibrium. However, the exergy destruction associated with chemical reactions will always be very high. The $E_{D}$ associated with heat transfer depends on (1) the difference between the average thermodynamic temperatures of the combustion gases and the heated fluid, and (2) the temperature level at which the heat transfer takes place. It should be noted that four causes of $E_{D}$ are considered in this case by following the procedure in Reference [39]. All these processes occur in real processes simultaneously and not successively.

According to the mentioned considerations, $E_{D}^{UN}$ was calculated under the assumptions that (1) no pressure drop occurs during combustion, (2) the combustion is stoichiometric (λ = 1) in order to minimize the exergy destruction due to chemical reactions, even though this would increase the adiabatic combustion temperature, and, thereby, the heat transfer inefficiencies, and (3) the minimum temperature difference for the heat transfer is just unattainable. Incidentally, the composition of the combustion gases in the unavoidable conditions would be different from that in the real case.

Table 2 summarizes the above points.

Along with the above conditions, it is worth highlighting that the $E_{D,k}^{UN}$ for each component must be calculated dynamically in every time step. Then, different $E_{D,k}^{UN}$ values would emerge while the productive conditions vary.

### 3.6. Endogenous and Exogenous Exergy Destructions 

The endogenous exergy destruction of the *k*th component ($E_{D,k}^{EN}$) reflects that part of the overall $E_{D}$ is exclusively related to the intrinsic irreversibilities within the component itself.

As discussed in Reference [34], various methods can be used for splitting the exergy destruction within system components into its endogenous and exogenous parts. Some of these approaches need simulation and assessment of theoretical thermodynamic cycles. Therefore, one main disadvantage of these methods is the complexity associated with the definition and the simulation of theoretical cycles for some systems [40]. Others require several sensitivity analyses based on the exergy balances and the fuel and product exergies [41]. Besides considerable effort required for doing mathematical calculations, these approaches also fail to split exergy destruction in dissipative components. The main drawback of the above-mentioned approaches is non-standard simulations required for the idealization of the plant based on the second law of thermodynamics that cannot be easily conducted by using available commercial software [13]. Nonetheless, a new straightforward and time-saving methodology has recently been developed [8] based on a systematic approach derived from a general process design and synthesis principles. Consequently, this methodology will be summarized and implemented below.

This methodological decomposition method is based on the idea that the exergy concept is independent from the whole structure, so that $E_{D,k}^{EN}$ in every component can be individually evaluated, according to its characteristics. In this way, the facility can be divided into reversible and irreversible subgroups. The term $E_{D,k}^{EN}$ results from keeping the *k*th component operating at its current exergetic efficiency ($\varepsilon_{k}$) while all the remaining components are assumed to be totally reversible, i.e., exhibiting an exergetic efficiency of 100% [8]. Notwithstanding these simplifications, it must be noted that the “idealization” of the remaining components should not change the organization and the structure of the flowsheet. Likewise, the overall product stream(s) must remain the same as in the real facility. As a result, it is possible to determine the productive contribution and the inefficiencies associated with different components.

The graphical representation of this approach is depicted in Figure 3. The upper part of the figure shows the facility under study where the real incoming resources and outgoing final products are displayed. The other four pictures illustrate how the endogenous exergy destruction in components S, V2, HC, and HX are calculated, while their $\varepsilon_{k}$ as well as the outgoing final product stream(s) (red arrows) are maintained the same as in the experimental facility.

Once the endogenous exergy destruction of the *k*th component is determined, the exogenous part can be calculated by subtracting this value from the real $E_{D,k},$ as shown in the following equation.
(3)$$E_{D,k}^{EX}=E_{D,k}-E_{D,k}^{EN}$$

Accordingly, $E_{D,k}^{EX}$ represents part of the exergy destruction in component *k* that exists because of the inefficient operation of the remaining components in the given structure of the system.

Application of the method to the experimental building thermal facility.

To begin with the calculation of $E_{D,k}^{EN}$ in our case study, the product of each component $E_{P,k}$ needs to be defined first. That product expresses the reason for owning and operating the component being considered under real conditions (i.e., with irreversibilities), while considering the rest of the system under a hypothetical ideal operation (i.e., without irreversibilities). In this regard, attention to the following issues is essential for the analysis of this building’s thermal facility.
DHW is one of the outgoing products of the facility supplied by the storage tank T located at the end of the production chain. That means that the conditions of DHW exclusively depend on the storage thermal conditions instead of the activation or deactivation of the other components. Two different situations are considered in this scenario as examples to make that clear. In one case, the requested DHW is entirely covered by the tank discharge, while, in another situation, there might be no demand for DHW. However, because of the temperature control strategy of the tank, the heat generation units are turned on to achieve the set-point temperature within the tank. Hence, the DHW demand should only be considered as the product of the tank $\left(E_{P,T}=E_{DHW}\right)$ whereas, for the remaining components of the system, the product (or part of the product) is the flow of thermal energy supplied to the tank $\left(E_{F,T}\right),$ as shown in Table 3.The product associated with the heating demand needs to be carefully considered. One might think that the exergetic value of the heating demand is the exergy of the flow of thermal energy at the surface temperature of the fan coil. Nonetheless, the exergy of the delivered heat has to be calculated at room air temperature (≈20°) since the purpose of a heating device in a real facility is to temper the room air, in order to satisfy the comfort conditions. Therefore, the heating exergy demand is much lower than the supplied exergy by the fan coil, and, hence, the endogenous exergy destruction in this component ($E_{D,FC}^{EN}$) is very high. This high endogenous exergy destruction within a downstream component of the system needs to be compensated by upstream ones, and, therefore, it results in high exogenous exergy destructions in the rest of the system.

Some components of the system, such as the condensing boiler inlet temperature fixing (ITF) and pumps, do not have productive purposes. Those components aim to satisfy some specific conditions for the boiler in order to provide the required mass flow by balancing the pressure losses, respectively. They are merely used just to enable the operation of the facility. Hence, all their $E_{D}$ would correspond to the exogenous part, since they are entirely caused by the operation of other components in the system.

#### 3.6.1. Binary Exogenous Exergy Destructions

Once the exogenous exergy destruction ($E_{D,k}^{EX}$) within every component is calculated, it can be split into several parts with each one being generated by a different component in the system. That is done by combining scenarios of two components working with their real exergetic efficiency ε while all other components operate reversibly with an exergetic efficiency equal to 100%. In this situation, the exergy destruction within the *k*th component (${E^{\prime }}_{D,k}$) consists of two parts: the endogenous exergy destruction that has already been calculated and the exogenous one caused only by the irreversibilities within component *i* ($E_{D,i\to k}^{EX}$). The value of the latter is obtained by subtracting $E_{D,k}^{EN}$ from the new exergy destruction of the *k*th component estimated in this new stage, as given by Equation (4).
(4)$$\Delta E_{D,i\to k}^{\mathrm{EX}}=E_{D,k}^{\prime }-E_{D,k}^{EN}$$

Figure 4 visualizes the methodology to obtain the binary exogenous exergy destruction caused within component *k* by the irreversibilities within component *i*.

The analysis of the binary interactions between upstream and downstream components of the energy chain reveals that the latter are the ones which mainly cause $E_{D}$ in the upstream components. In fact, if there is no energy recirculation within the system being considered, the exogenous part of exergy destruction within component *i* due to thermodynamic inefficiencies occurring in the upstream component *k* ($E_{D,k\to i}^{EX}$) would be zero because, in this case, ${E^{\prime }}_{D,i}$ would be exactly the same as the $E_{D,i}^{EN}$ (see Figure 5).

#### 3.6.2. Mexogenous Exergy Destructions

Besides endogenous and binary exogenous causes of exergy destruction within component *k*, there is another source of exergy destruction in this component due to the simultaneous interactions between the *k*th component and the rest of the system operating under its real efficiency. This is called mexogenous (i.e., mixed exogenous) exergy destruction [14] and is calculated by subtracting the sum of the binary exogenous exergy destructions from the total exogenous exergy destruction within the *k*th component, as shown in the following equation.
(5)$$E_{D,K}^{MEX}=E_{D,K}^{EX}-\overset{N}{\underset{\begin{array}{c} i\\ i\neq k \end{array}}{\sum }}E_{D,i\to k}^{\mathrm{EX}}$$

#### 3.6.3. Considering Real Characteristic Curves

The decomposition method offers an appealing approach to obtain the endogenous exergy destruction within different components of a system since the required mathematical effort and calculation time are significantly lower than those by other methods. However, this approach deals only with exergetic variables (such as $E_{F,k}$, $E_{P,k},$ and $\varepsilon_{k}$), which should remain constant, without considering the real physical characteristics of the components under different operating conditions. Therefore, the approach could lead to some controversial results. For instance, when only one component of the system is operating under its real conditions (with its real $\varepsilon_{k}$) and the rest of the system is assumed to be ideal (ε=1), the virtual product that this component is supposed to supply $\left(E_{P,k}\right)$ is inconsistent (e.g., outside of the component operating range, for example, too low demand for CB that is not feasible). Moreover, the ε of the component under different operating conditions does not remain the same.

Consequently, a similar standpoint can be carried out if, instead of working with a constant exergetic efficiency, the real one under the new thermodynamic conditions (${\varepsilon^{\prime }}_{k}$) is considered, based on the characteristic curves of each component. The endogenous exergy destruction obtained from this approach ($E_{D,k}^{EN^{*}}$) represents the authentic $E_{D}$ that a component would have, when supplying the corresponding $E_{P,k}$.

The difference between the endogenous exergy destruction based on the real characteristic curves of the components ($E_{D,k}^{EN^{*}}$) and the one obtained according to the decomposition method ($E_{D,k}^{EN}$) refers to the irreversibilities due to the fact that the component needs to adapt itself to the new thermodynamic conditions $\left({\varepsilon^{\prime }}_{k}\right)$.
(6)$$\Delta E_{D,k}^{EN}=E_{D,k}^{EN^{*}}-E_{D,k}^{EN}$$

### 3.7. Combination of the UN/AV and EN/EX Parts of Exergy Destruction

Overviewing the general DAEA developed so far, $E_{D}$ in the *k*th component of the system has been divided into its avoidable and unavoidable parts and into its endogenous and exogenous sections in the present paper. Those parts of exergy destruction can be combined to calculate the following four variables: (1) $E_{D,k}^{AV.EN}$ is the exergy destruction that can be reduced by the improvement of the *k*th component itself, (2) $E_{D,k}^{AV.EX}$ is the exergy destruction within the *k*th component, which can be decreased by enhancing the performance of the other components, (3) $E_{D,k}^{UN.EN}$ is the unavoidable exergy destruction, which corresponds to the inherent limitations of the component being considered, and (4) $E_{D,k}^{UN.EX}$ is the unavoidable exergy destruction that embodies the structural constraints and component interactions. They are calculated as follows.

## 4. Numerical Values and Results

Down below, the numerical results of the analysis are summarized.
(7)$$E_{D,k}^{AV\cdot EX}=\frac{E_{D,k}^{AV}\cdot E_{D,k}^{EX}}{E_{D,k}}$$
(8)$$E_{D,k}^{UN\cdot EN}=\frac{E_{D,k}^{UN}\cdot E_{D,k}^{EN}}{E_{D,k}}$$
(9)$$E_{D,k}^{UN\cdot EX}=\frac{E_{D,k}^{UN}\cdot E_{D,k}^{EX}}{E_{D,k}}$$
(10)$$E_{D,k}^{UN\cdot EN}=\frac{E_{D,k}^{UN}\cdot E_{D,k}^{EN}}{E_{D,k}}$$

### 4.1. Characterization of the Components

As previously mentioned, the LQCB experimental facility was used in a four-day test. In the present analysis, 18 thermocouples (with an uncertainty of ±0.15 °C) and seven flowmeters (with an accuracy of ±0.1%) were used.

The DHW and heating demands were calculated with TRNSYS v17, and then the control system was accordingly programmed. Those demands are illustrated in Figure 6. Although the data were acquired every 10 s, the dynamic mathematical model was built based on a 5-min time step, since that time step was considered sufficient for accurately representing the transient start-up and shut-down of the components.

As previously discussed, a mathematical characterization equation of every component has been obtained. As an example, the calculation of the output variables of the component C ($T_{2_{Dep}}^{Calc},\;T_{5_{Dep}}^{Calc}$, and $T_{6_{Dep}}^{Calc}$) is demonstrated. The variables obtained as TRNSYS outputs $\left(T_{i_{Dep}}^{Tr}\right)$ are corrected to the actual experimental facility measurements by means of the MATLAB System Identification Toolbox, which, in this case, becomes the following.
(11)T2DepCalc(t)=T7IndRe(t)
(12)T3DepCalc(t)={T7IndRe(t)·0.9649m˙CB=0T3DepTr(t)m˙CB≥ 0
(13)T6DepCalc(t)=m˙S(t)·T1IndRe(t)+m˙CB(t)·T4IndRe(t)m˙S(t)+m˙CB

Accordingly, it can be seen how the effects of the activation and deactivation of the pumps and the hydraulic compensator inertia are taken into account (see the condition on ${\dot{m}}_{CB}$ in Equation (12)).

The relative error between the experimental data and the simulation results shows the difference among the real dependent values and the dependent values calculated from the TRNSYS and MATLAB combination $\left(T_{i_{Dep}}^{Calc}-T_{i_{Dep}}^{Re}\right)$ is less that ±3% for each individual component of the facility. However, when all the subsystems are linked, the maximum error for the total facility is lower than ±9%. This is larger than the error for the individual components because the uncertainty increases when all the components are modelled simultaneously.

### 4.2. Conventional Exergy Analysis

A conventional exergetic analysis calculates the exergy destruction within each component, $E_{D,k}$, in every time step.

Figure 7 illustrates the total exergy destruction in the four-day period ($E_{D}^{TOT}=1259\;\mathrm{kWh}$) as well as the percentages of the contribution of exergy destruction within different components to the total exergy destruction. The causes of $E_{D,k}$ in all components are individually justified in Section 3.5 and summarized in Table 4.

As expected, components including a combustion process have, by far, the highest exergy destructions. Due to the previously explained control strategy, CB works as a backup device and its operating hours are, thus, less than S. However, 46% of the overall exergy destruction occurs in CB and only 27% in S. One reason that $E_{D,S}$ is much lower than $E_{D,CB}$ is because S generates heat and electricity simultaneously, so that its exergetic efficiency is much higher than $\varepsilon_{CB}$. Another reason is that CB has a higher nominal capacity than S.

Following the instructions written in Section 3.5, for the components CB and S, the exergy destructions are separated into the following groups, according to their causes: (a) friction, (b) mixing, (c) chemical reactions, and (d) heat transfer. The results show that almost all the exergy destructions belong to the last two categories: 59.7% and 64.3% are caused by chemical reactions in S and CB (comb), respectively, whereas 40.2% and 35.6% are due to heat transfer (HT) in S and CB, respectively (see Figure 8).

### 4.3. Unavoidable/Avoidable $E_{D}$

The unavoidable and avoidable $E_{D}$ of each component were dynamically calculated following the guidelines outlined above and the average percentage results are presented in Figure 9.

As can be seen, the $E_{D}^{UN}$ values within the most important components are much larger than the avoidable ones ($E_{D}^{AV}$). For example, in S and CB, approximately 88% of the exergy destruction is unavoidable because, when the destructions associated with chemical reactions are reduced, the internal heat transfer irreversibilities increase due to the rise of the combustion temperature.

The average results of the conventional exergy analysis together with the DAEA are listed in Table 4.

As a consequence of the facility’s physical model, the values of $E_{D}$ decrease significantly after the HC component. The irreversibilities within the downstream components decrease plenty once the energy passes through the HC component. This can be easily explained because, after that unit, the downstream components are split into the DHW branch and the heating branch. Therefore, they are dealing with a lower flow of energy (and exergy). Therefore, less exergy destruction takes place within the downstream components.

### 4.4. Endogenous/Exogenous $E_{D}$

The contribution of endogenous and exogenous exergy destructions within different components for the four-day test are illustrated in Figure 10.

As noted above, the closer the components are to the final products, the lower the amount of their exogenous exergy destruction is. In other words, due to the fact that the main product of the system remains constant $\left(E_{P,TOT}=E_{DHW}+E_{Heat}\right)$, if the downstream components cause additional exergy destruction, the upstream components need to produce a larger output to offset those additional irreversibilities. This situation results in a larger $E_{D}$ in the respective upstream components. This is the main reason that explains the high value of $E_{D}^{EX}$ for both the CB and the S components.

Having no productive purpose, the exergy destruction in ITF is entirely exogenous. On the contrary, the exergy destruction in the heat exchanger (HX), storage tank (T), and fan coil (FC) are entirely endogenous because those components are directly linked to the final product of the overall system.

The exogenous exergy destruction in every component can be split to consider the contributions from each component by using the binary interrelation approach. Figure 11 displays those binary interactions in terms of exergy destruction, which is colored according to the component associated with its origin.

As expected, FC is the component with a noticeable contribution to the exogenous $E_{D}$ in the upstream components, not only because it is located at the end of the heating branch, but also because of its very large endogenous exergy destruction.

### 4.5. Considering Real Characteristic Curves

A step forward in the application of DAEA is the calculation of endogenous exergy destruction within every component based on the real exergetic efficiency of this component $\left({\varepsilon^{\prime }}_{k}\right)$ instead of having a constant one, where the thermodynamic state of the system changes at every time step. The difference between the endogenous exergy destruction obtained from the previously mentioned method and the one calculated based on the decomposition approach ($\Delta E_{D,k}^{EN}$) indicated that, in some cases, dealing with constant ε for the calculation of $E_{D,k}^{EN}$ is a simplified assumption that might lead to controversial results.

Figure 12 compares the values of $E_{D,k}^{EN}$ and $E_{D,k}^{EN^{*}}$ for different components of the system.

The major difference is found in the condensing boiler because its real exergetic efficiency curve, ${\varepsilon^{\prime }}_{CB}$, is not flat and decreases significantly when the demand decreases. Other components with relatively large $\Delta E_{D,k}^{EN}$ are the supply and return collectors (C) and the hydraulic compensator (HC) because their endogenous exergy destructions, caused mainly by mixing of fluids with different temperatures, changes significantly at different operating conditions. For the remaining components, the endogenous exergy destruction is similar in both methods.

### 4.6. Combination of the UN/AV, EN/EX Parts of Exergy Destruction

Lastly, new findings can be identified when unavoidable and avoidable exergy destructions are combined with the endogenous and exogenous exergy destructions. Detailed results from the DAEA application to the experimental building’s thermal facility for a period of four days are given in Table 5. The upgrading prospective of each component can be detected there.

Similarly, Figure 13 summarizes the contributions of the four possible combinations to the entire system.

It can be seen that more than three-quarters of the overall exergy destruction is unavoidable, with 39% of it being due to the components’ internal limitations, while the remaining 61% is due to the interactions among components and to system structural restrictions.

Concerning the avoidable exergy destruction, only 8% of the total $E_{D}$ can be reduced by using the best technological alternatives currently available in the market, whereas 15% of the global exergy destruction can be avoided by improving the interrelations among components of the system by customizing and improving the control system.

In summary, according to the results of the DAEA, the improvement potential of the overall system is rather low and limited. Likewise, the fact that $E_{D}^{AV.EX}$ is higher than $E_{D}^{AV.EN}$ proves that components of the system are strongly interconnected. Thus, a better control strategy can improve the overall efficiency of the system. Nevertheless, it must be noted that $E_{D}^{AV.EX}$ considers both the interconnections between components as well as the imperfections coming from the others because they are not using the best equipment available in the current market.

As a whole, DAEA analysis gives good insight on the optimisation possibilities. On the other hand, a heating system based on a heat pump with a floor heating system would have a much better performance compared to the described Stirling engine/gas boiler and fain coil unit system. Even so, this comparison is not within the frame of this study, but further analysis can be done for project design optimization.

Therefore, the obtained information can be used for control-strategy optimization, a fault detection approach, or even design optimization.

## 5. Discussion

A Dynamic Advanced Exergy Analysis (DAEA) is applied to a building thermal energy system. Buildings are major contributors to the primary energy demand so that the awareness of their improvement potential is an essential requirement for reducing energy demand and ${\mathrm{CO}}_{2}$ emissions. That assessment can be made through a DAEA application, as presented in this article.

A DAEA allows engineers to determine which inefficiencies could be avoided in the current technical limitations by splitting the exergy destruction into avoidable and unavoidable parts. Moreover, a DAEA, using the concepts of endogenous and exogenous exergy destructions, identifies the inefficiencies caused by the component itself as well as those coming from the imperfections of the remaining components. Consequently, it provides very useful information that cannot be supplied through a conventional exergetic analysis.

Nevertheless, even with the suitability of that analysis, some shortcomings need to be mentioned. First, the calculation of unavoidable exergy destruction is associated with some subjectivity during its estimation. Even so, this standpoint depends on the engineers’ judgment. Therefore, accordingly, the chosen criteria are explained in detail and justified throughout this article.

Furthermore, when endogenous and exogenous $E_{D}$ are calculated, some information might be lost if some real thermodynamic quantities, such as a varying exergetic efficiency, are not considered. However, this can be overcome by the incorporation of the real characteristic curves of the components.

Lastly, apart from the insight that a detailed distribution of the exergy destruction $E_{D}$ can provide, one of the advantages in the application of DAEA is that there is no need to predefine any reference conditions to evaluate the performance of a system. Hence, it is a generic method that can be used for control, diagnosis, or even design purposes. In addition to that, since exergy destruction is employed as the base parameter, the real involvement of every component within the overall system is regarded. This implication should not be noticed if exergetic efficiency (ε) would have been taken instead. Even if the upstream components have higher efficiencies than the downstream ones, the exergy destruction is much higher in the group of the former than the latter, due to the bigger amount of exergy those components deal with. That is the case, for example, of the supply and return collector (C) and the fan coil (FC). As a result, some information can be misinterpreted if the $E_{D}$ is not used.

Moreover, one of the innovative features of this paper is associated with the dynamism used in the DAEA, since a steady state can never be related to building energy supply facilities. Therefore, before any calculation, the development of a reliable dynamic model for every component and for the entire facility is of crucial importance. This was achieved on the basis of recorded data from an experimental plant and the grey-box modeling, by the combination of TRNSYS models and MATLAB System Identification Toolbox for the parameter adjustment, with an overall error below ±9%.

Despite that, certain issues arise when an AEA is dynamically applied. In the first instance, since thermal parameters are continuously time-varying, special care needs to be given to the Fuel and Product definition for every component. In addition, the inertia over the system plays a relevant role, which must be taken into account. Furthermore, the dynamic operating conditions also introduce non-flat exergetic efficiency curves. However, those complexities have been overcome during the development of this paper.

## 6. Conclusions

To conclude, a DAEA elaborately resolves the way to allocate the inefficiencies of a system within its components, by considering both the internal irreversibilities as well as the interconnections between components. Due to the dynamic character, its application in building facilities seems to be more complicated than in systems where a steady state operation can be assumed. Despite that, a DAEA can be equally applied by merely adjusting some assumptions. Therefore, a step forward on the reduction of the energy use can be achieved by using the information provided by this analysis.

The future steps should include the economic quantification of those results as well as the environmental impact calculation or the translation from the exergetic units to the monetary ones and to units of environmental impacts. That can be performed with the aid of exergoeconomic and exergoenvironmental analysis.

## Figures and Tables

**Figure 1 entropy-22-00032-f001:**
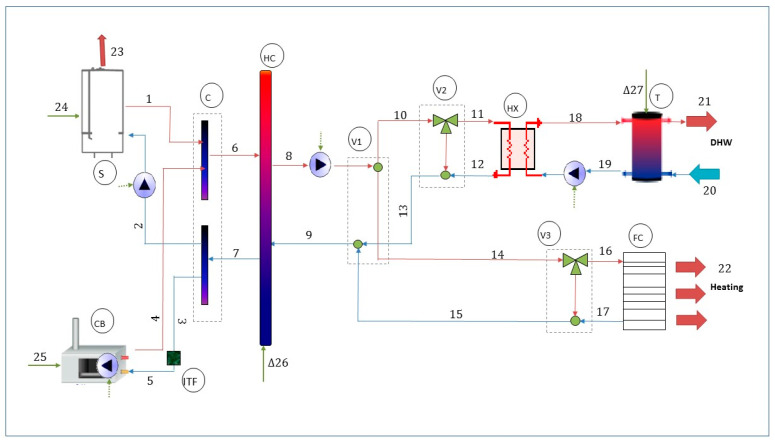
Scheme of the building thermal facility under study.

**Figure 2 entropy-22-00032-f002:**
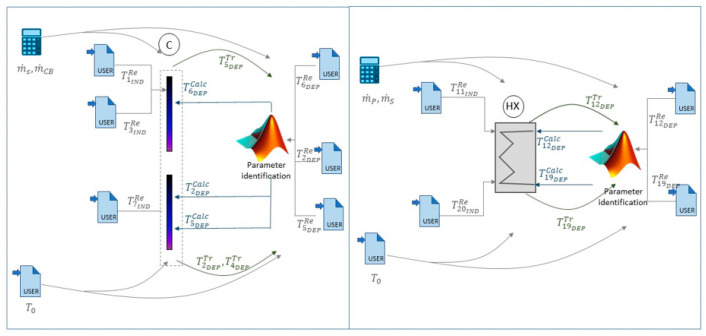
Representative image for the C and HX characterization through the Simulation Studio interface of TRNSYS.

**Figure 3 entropy-22-00032-f003:**
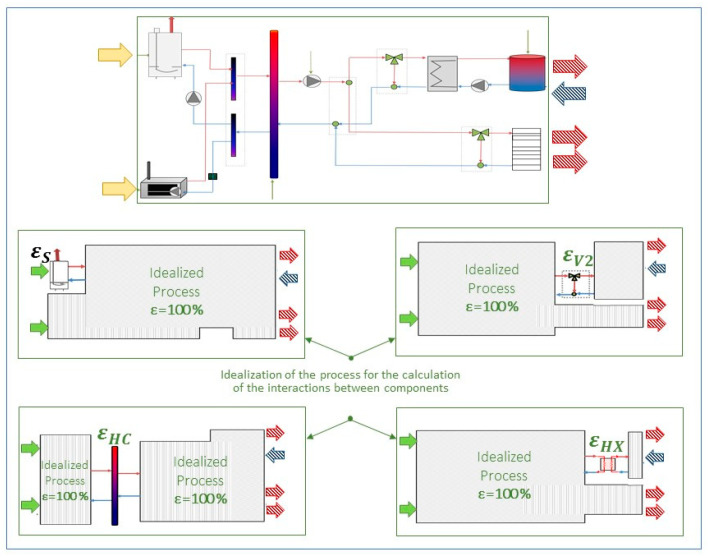
Decomposition method for calculating the $E_{D}^{EN}$, adapted from Reference [35].

**Figure 4 entropy-22-00032-f004:**
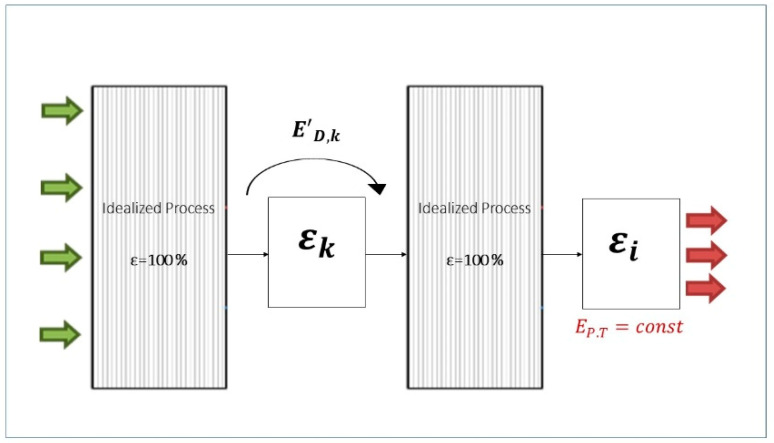
Graphical explanation of the methodology to obtain $E_{D,i\to k}^{EX}$.

**Figure 5 entropy-22-00032-f005:**
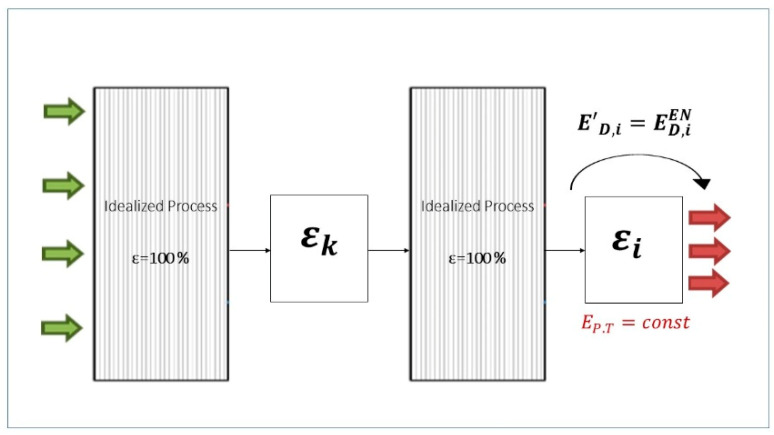
Graphical justification for ${E^{\prime }}_{D,i}=E_{D,i}^{EN}$ in the downstream component i.

**Figure 6 entropy-22-00032-f006:**
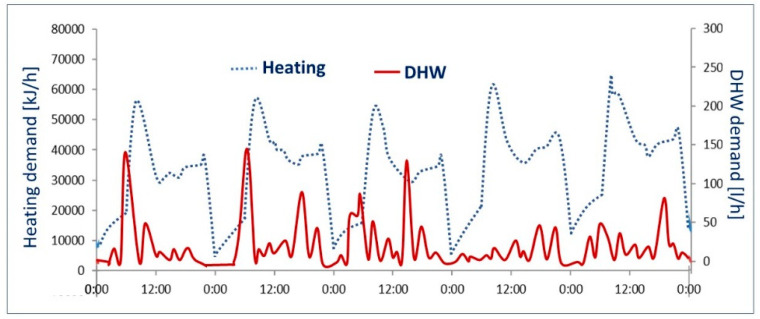
Heating and DHW demand covered by the facility.

**Figure 7 entropy-22-00032-f007:**
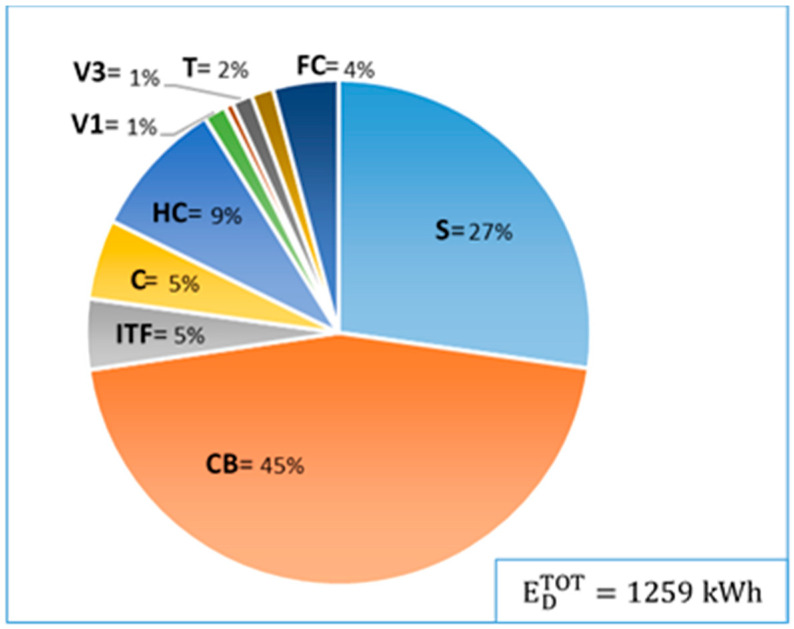
Average contribution of $E_{D,k}$ in each component to the overall exergy destruction.

**Figure 8 entropy-22-00032-f008:**
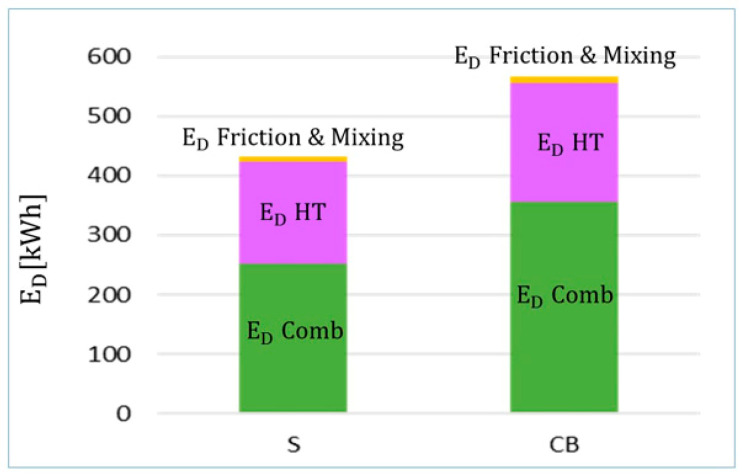
Exergy destruction distribution for the combustion devices based on Reference [37].

**Figure 9 entropy-22-00032-f009:**
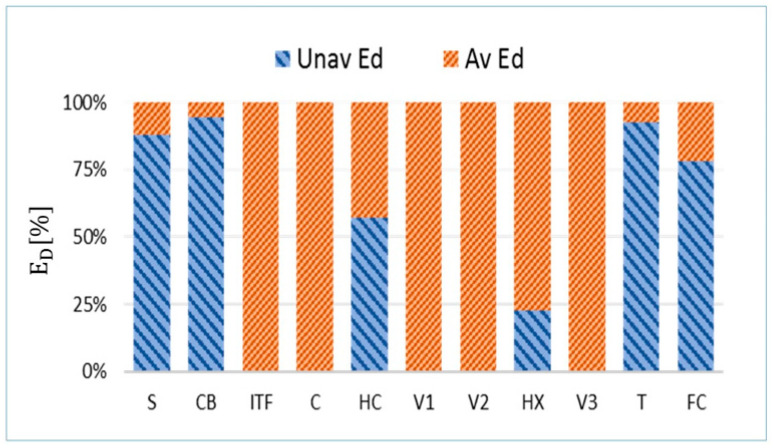
Percentage of unavoidable and avoidable exergy destructions in each component of the experimental facility.

**Figure 10 entropy-22-00032-f010:**
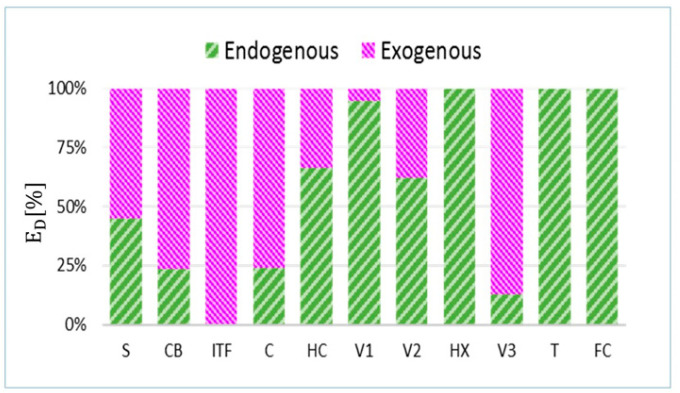
Share of endogenous and exogenous exergy destructions in each component.

**Figure 11 entropy-22-00032-f011:**
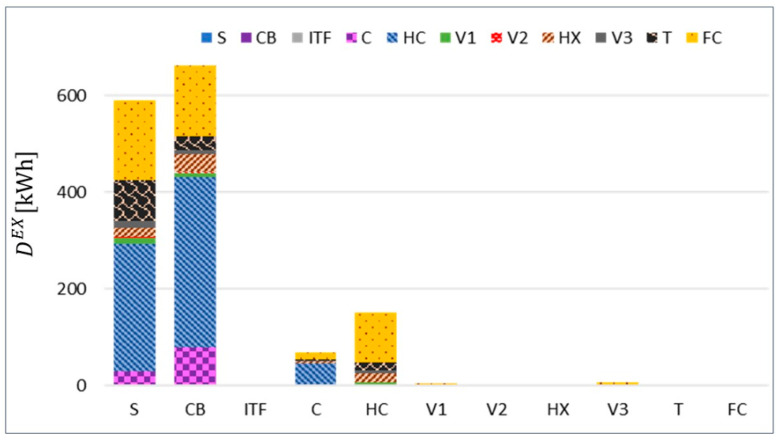
Average binary exogenous exergy destruction of every component.

**Figure 12 entropy-22-00032-f012:**
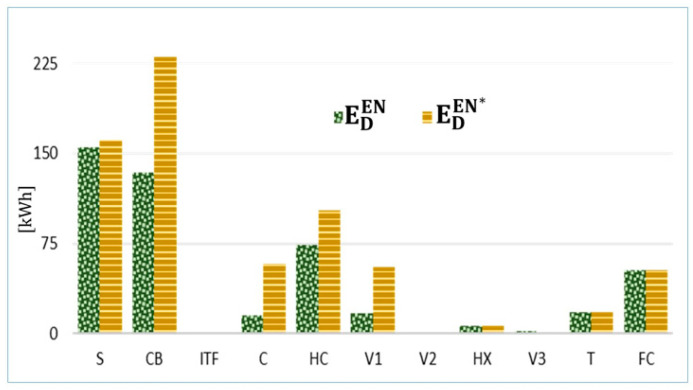
Endogenous exergy destruction of every component obtained from the decomposition method ($E_{D,k}^{EN}$) and from the real characteristic curves ($E_{D,k}^{EN^{*}}$).

**Figure 13 entropy-22-00032-f013:**
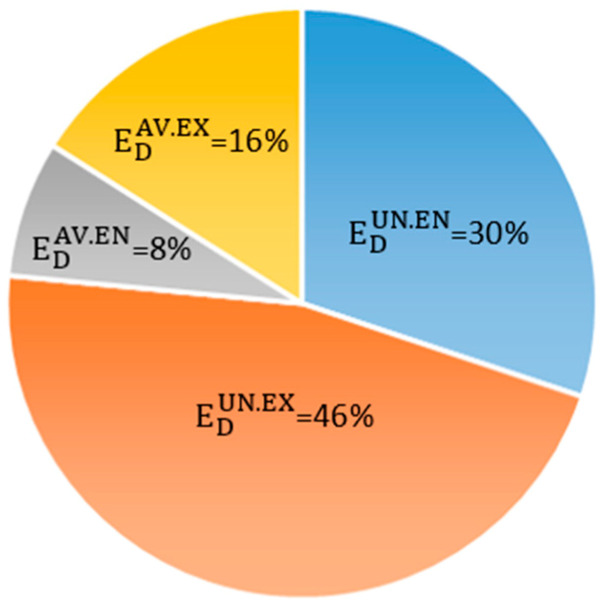
Contributions of $E_{D}^{UN.EN}$, $E_{D}^{UN.EX}$, $E_{D}^{AV.EN}$, and $E_{D}^{AV.EX}$ to the total exergy destruction in the entire system.

**Table 1 entropy-22-00032-t001:** List of components used for DAEA.

Name	Description
S	Micro-cogeneration Stirling engine
CB	Condensing boiler
ITF	CB inlet temperature fixing
C	Supply and return collectors
HC	Hydraulic compensator
V1	DHW and heating mixer and splitter
V2	HX mixer and splitter
HX	Heat exchanger
V3	Heating mixer and splitter
T	Storage tank
FC	Fan coil

**Table 2 entropy-22-00032-t002:** Justification and achievement of the unavoidable exergy destruction in every component.

n.	CAUSES for $E_{D}$	UNAVOIDABLE $E_{D}$ Achievement
xC, V1, V2, V3	xMixing with different states	xMixing at equal temperatures and pressures are assumed
xITF	xMixing for achieving the required conditions	xFlow enters with the same thermodynamic conditions as required CB’s inlet
xFC	xThermal and pressure losses	xHighest energetic efficiency + no pressure losses
xHX	xTemperature difference, pressure drop, and thermal losses	xMinimum ΔT of the average temperatures + adiabatic HX + no pressure losses + constant and maximum available heat transfer coefficient
xHC, T	xMixing, tank average T, charging rate, heat losses to the environment	xInsulation addition to the tank boundaries + no pressure losses
S, CB	Friction, mixing, chemical reaction, heat transfer	Stoichiometric combustion (λ = 1) + minimum ΔTª for the heat transfer + no pressure losses

**Table 3 entropy-22-00032-t003:** Description of the components and their final objective product used for calculating the $E_{D,k}^{EN}$.

k	$E_{P,k}$ for $E_{D,k}^{\mathrm{EN}}$ Calculation
S	$\left(E_{F,T}+E_{Heat}\right)\cdot \%E_{P,S}$
CB	$\left(E_{F,T}+E_{Heat}\right)\cdot \%E_{P,CB}$
ITF	Not applicable
C	$E_{F,T}+E_{Heat}$
HC	$E_{F,T}+E_{Heat}$
V1	$E_{F,T}+E_{Heat}$
V2	$E_{F,T}$
HX	$E_{F,T}$
V3	$E_{Heat}$
T	$E_{DHW}$
FC	$E_{Heat}$

$E_{Heat}$, Heating exergy demand; $E_{DHW}$, DHW exergy demand; $E_{F,T}$, Exergy of the heating stream supplied to the tank; $\%E_{P,S}$, The share of S in satisfying the demand; $\%E_{P,CB}$, The share of CB in satisfying the demand.

**Table 4 entropy-22-00032-t004:** Average results of the DAEA and the conventional exergy analysis.

Component *k*	$E_{F,k}$ [kWh]	$E_{P,k}$ [kWh]	$\varepsilon_{k}$ [%]	$E_{D,k}$ [kWh]	$E_{D,k}^{\mathrm{UN}}$ [kWh]	$E_{D,k}^{\mathrm{AV}}$ [kWh]
S	456.01	111.77	25%	344.24	302.67	41.56
CB	665.36	91.44	14%	573.92	542.62	31.30
ITF	116.75	59.31	51%	57.44	0.00	57.44
C	180.30	115.29	64%	65.01	0.00	65.01
HC	146.93	35.01	24%	111.92	63.99	47.93
V1	94.39	76.61	81%	17.78	0.00	17.78
V2	234.41	233.74	100%	0.68	0.00	0.68
HX	29.24	22.78	78%	6.46	1.47	4.99
V3	84.80	68.50	81%	16.30	0.00	16.30
T	32.77	14.97	46%	17.80	16.49	1.31
FC	65.33	12.34	19%	53.00	41.32	11.68

**Table 5 entropy-22-00032-t005:** Detailed aggregated results obtained from the DAEA.

Component *k*	$E_{D}$ [kWh]	Unavoidable		Avoidable	
		$E_{D}^{\mathrm{UN},\mathrm{EN}}$ [kWh]	$E_{D}^{\mathrm{UN},\mathrm{EX}}$ [kWh]	$E_{D}^{\mathrm{AV},\mathrm{EN}}$ [kWh]	$E_{D}^{\mathrm{AV},\mathrm{EX}}$ [kWh]
S	302.67	149.25	153.43	5.67	35.96
CB	542.62	125.39	417.23	8.41	22.28
ITF	0.00	0.00	0.00	0.00	57.16
C	0.00	0.00	0.00	15.36	49.13
HC	63.99	48.41	15.58	25.58	22.29
V1	0.00	0.00	0.00	16.87	0.91
V2	0.00	0.00	0.00	0.40	0.25
HX	1.47	1.49	0.00	4.97	0.00
V3	0.00	0.00	0.00	2.11	14.21
T	16.49	16.13	0.00	1.67	0.00
FC	41.32	39.55	0.00	13.44	0.00

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
