# Peer review of "Advanced Exergy Analysis in the Dynamic Framework for Assessing Building Thermal Systems"

_entropy, 2019, doi:10.3390/e22010032_

Round 1
Reviewer 1 Report
The paper on „Advanced Exergy Analysis in the Dynamic Framework for Assessing Building Thermal Systems“ describes the application of the so-called “Dynamic Advanced Exergy Analysis” to a building heating system satisfying the space heating and domestic hot water demand for three single family home in northern Spain. The aim of the study was to identify sources for exergy destruction in the presented system and give suggestions for improvements of the situation. To improve the modelling of the system measured values from a laboratory facility have been used to tune the models.
The authors give a good and understandable overview over the used method and how to apply it. On the other hand there are some remarks:
- The authors are stating that dynamic exergy analyses have not been applied before to building energy systems. Just to give one example, already during the course of the IEA ECBCS Annex 49 project on “Low Exergy Systems for High-Performance Buildings and Communities” many papers describing a dynamic exergy analysis have been presented. So, the authors’ statement should be relativized.
- It is quite hard to understand the analysed system layout, since there are descriptions on Figure 1 missing. Here some better explanations should be added.
- How are the demand profiles for space heating and domestic hot water preparation generated? What tapping cycles and climate conditions (TRY?) are used? How are these profile used in the test facility?
- For all exergy analyses the choice of the reference temperature is a crucial and important part. Some more information how this choice is made in the presented paper would increase the quality of the paper.
- The results show the expected outcome that combustion processes are having a great impact on the overall exergy destruction of a heating system. Anyhow, the analysis gives a good insight on the optimisation possibilities. On the other hand a heating system based on a heat pump with a floor heating system would have a much better performance compared to the described Stirling engine/gas boiler and fain coil unit system. Even so this comparison was not within the frame of this study, short information about this fact would have been great.
Author Response
"Please see the attachment."

Reviewer 2 Report
The paper "Advanced Exergy Analysis in the Dynamic Framework for Assessing Building Thermal Systems" has an interesting and significant content. The quality of presentation is good enough, however, there are some points that could be improved from the reviewer point of view. So the following suggestions are summarized:
1) In the title, change the word "assesing" to "assessing".
2) In the introduction, as it is now it emphasizes so much about the "advanced energy analysis" which can be found in many works delivered by the author "G. Tsatsaronis" and colleges. So it would be good to include something related to the state of the art in methods or techniques to evaluate the efficiency of building facilities. This is suggested to highlight more precisely the contribution to the topic.
3) There are many errors of references through the paper: line 153, 166, 199, 219, 228, 238, 246, 249, 296, 363, 395, 421, 450, 451, 460, 522,532, 549, 554, 564, 576 and so on.
4) In the literature there are a method which is something related to the Mexogenous Exergy Destruction which is called "The Encounter Effect Method" used in the diagnosis method. Ref.:
Thermoeconomic diagnosis of large industrial boilers: microscopic representation of the exergy cost theory
VHR Hernández, AV Capilla, L Carlos, C Uson - 2005what would be the difference between the method proposed by the authors and the one found in the reference given above?
5)
Author Response
"Please see the attachment."

Reviewer 3 Report
The article "ADVANCED EXERGY ANALYSIS IN THE 2 DYNAMIC FRAMEWORK FOR ASSESING 3 BUILDING THERMAL SYSTEM" proposes an application of a novel method to carry out exergy analysis.
The article, indeed, is quite good. Nevertheless, the focus of the report became the application of the method, whereas the explanation of the advanced exergy analysis was neglected. One could think that is an application of the "traditional" exergy analysis taking into account the transient conditions. I think this should be mandatory, and the authors must explain this new method better. Otherwise, the intent of the article is missed.
Regarding the quality of the figure, it must be improved; all figures are in low resolution. The references are not in the journal template form. In the abstract, there is not much reference to conclusions, and the outcomes itself is inadequate.
Authors could be more synthetic in their explanations and remember the reader from this interdisciplinary journal may not be a specialist in exergy analysis, therefore, it must be included more comments about this advanced exergy method.
Regarding the application, again, it is quite good. Nevertheless, it should be considered an improvement in the presentation of the data with an error and statistical analysis, since authors account for several experimental data.
Authors should improve the correlation of the article with the template, for instance, the nomenclature. All figures are in low resolution, please improve.
Author Response
"Please see the attachment."

Round 2
Reviewer 1 Report
No further comments
Reviewer 2 Report
After having reviewed the paper and seen that suggestions from reviewers have been considered, then I can suggest the paper for publication.
Good Job.
Reviewer 3 Report
The authors carried the reviews accordingly to reviewers. Therefore, it can be accepted